# Extracellular Vesicles Released after Doxorubicin Treatment in Rats Protect Cardiomyocytes from Oxidative Damage and Induce Pro-Inflammatory Gene Expression in Macrophages

**DOI:** 10.3390/ijms232113465

**Published:** 2022-11-03

**Authors:** Chontida Yarana, Panjaree Siwaponanan, Chayodom Maneechote, Thawatchai Khuanjing, Benjamin Ongnok, Nanthip Prathumsap, Siriporn C. Chattipakorn, Nipon Chattipakorn, Kovit Pattanapanyasat

**Affiliations:** 1Center for Research and Innovation, Faculty of Medical Technology, Mahidol University, 999 Phuttamonthon 4 Road, Salaya, Nakhon Pathom 73170, Thailand; 2Center of Excellence for Microparticle and Exosome in Diseases, Department of Research and Development, Faculty of Medicine Siriraj Hospital, Mahidol University, Bangkok 10700, Thailand; 3Cardiac Electrophysiology Research and Training Center, Faculty of Medicine, Chiang Mai University, Chiang Mai 50200, Thailand; 4Cardiac Electrophysiology Unit, Department of Physiology, Faculty of Medicine, Chiang Mai University, Chiang Mai 50200, Thailand; 5Center of Excellence in Cardiac Electrophysiology Research, Chiang Mai University, Chiang Mai 50200, Thailand; 6Department of Oral Biology and Diagnostic Sciences, Faculty of Dentistry, Chiang Mai University, Chiang Mai 50200, Thailand; 7Graduate Program in Immunology, Department of Immunology, Faculty of Medicine Siriraj Hospital, Mahidol University, Bangkok 10700, Thailand

**Keywords:** doxorubicin, cardiotoxicity, extracellular vesicle, oxidative stress, macrophage

## Abstract

Doxorubicin (DOXO)-induced cardiomyopathy (DIC) is a lethal complication in cancer patients. Major mechanisms of DIC involve oxidative stress in cardiomyocytes and hyperactivated immune response. Extracellular vesicles (EVs) mediate cell–cell communication during oxidative stress. However, functions of circulating EVs released after chronic DOXO exposure on cardiomyocytes and immune cells are still obscured. Herein, we developed a DIC in vivo model using male Wistar rats injected with 3 mg/kg DOXO for 6 doses within 30 days (18 mg/kg cumulative dose). One month after the last injection, the rats developed cardiotoxicity evidenced by increased BCL2-associated X protein and cleaved caspase-3 in heart tissues, along with N-terminal pro B-type natriuretic peptide in sera. Serum EVs were isolated by size exclusion chromatography. EV functions on H9c2 cardiomyocytes and NR8383 macrophages were evaluated. EVs from DOXO-treated rats (DOXO_EVs) attenuated ROS production via increased glutathione peroxidase-1 and catalase gene expression, and reduced hydrogen peroxide-induced cell death in cardiomyocytes. In contrast, DOXO_EVs induced ROS production, interleukin-6, and tumor necrosis factor-alpha, while suppressing arginase-1 gene expression in macrophages. These results suggested the pleiotropic roles of EVs against DIC, which highlight the potential role of EV-based therapy for DIC with a concern of its adverse effect on immune response.

## 1. Introduction

Chemotherapy and radiotherapy in cancer patients lead to an elevated risk of late-onset cardiovascular disease, a leading cause of death in cancer survivors [1]. In persons who had childhood cancer and received chemotherapy, the risk of cardiovascular disease-related death in adulthood is eight times higher than in those without childhood patients [2]. Due to the increasing amount of cancer survivors, late-onset cardiotoxicity is more evident and prompts greater clinician awareness [3]. Among chemotherapeutic agents, doxorubicin (DOXO) is the prototype anthracycline that causes cardiotoxicity. DOXO can induce cardiac cell death via apoptosis, autophagy, pyroptosis, and ferroptosis [4]. The major mechanisms of DOXO-induced cardiomyopathy (DIC) involve the accumulation of oxidative stress in heart tissues [5] and vascular walls [6], as well as an increase in systemic low-grade inflammation [7,8].

Extracellular vesicles (EVs) are nano-sized vesicles released from almost all types of cells under both physiological and pathological conditions [9]. As a cell–cell communication vector, EVs carry multiple biomolecules including nucleic acids, proteins, lipids, and even cell organelles such as mitochondria [10,11]. These are transferred from one cell to another [12,13], and modulate biological function of their recipient cells [14]. Under oxidative stress conditions, previous studies revealed that EVs can have both protective and harmful effects on cells. EVs can contain antioxidants that reduce oxidative damage to recipient cells [15,16]. However, in certain situations, EVs can transfer reactive oxygen species (ROS) and reactive nitrogen species, as well as oxide-producing agents that could further aggravate oxidative damage of the recipient cells [17,18].

A growing body of evidence has shown that chemotherapy induces greater EV release from both normal [8,19,20] and cancer cells [21]. EVs released from different sources of cells possess different functions. Exosomes from cardiomyocytes treated with DOXO activated macrophages to release both pro- and anti-inflammatory cytokines [22]. In contrast, exosomes from bone marrow- and adipose-derived mesenchymal stem cells protected DIC by inhibiting inflammatory responses and fibrosis [23]. EVs in circulation are heterogeneous in their tissue-cellular origin and have been proposed as a potential biomarker for DOXO-induced cardiomyopathy (DIC) [20,24]. However, their exact effects on normal cells after the cessation of DOXO treatment has never been clearly elucidated. In this study, we investigated the role of EVs derived from rat sera one month after completing DOXO treatment (18 mg/kg cumulative dose) on cardiomyocytes and macrophages. Results revealed that EVs effectively protected cardiomyocytes from oxidative stress insults via upregulating antioxidant enzyme gene expression. However, they induced macrophages pro-inflammatory responses by upregulating the interleukin-6 (*Il6*) and tumor necrosis factor-alpha (*Tnf*) genes. Our findings suggest a novel mechanism by which EVs protect cardiomyocytes from long-term oxidative stress damage due to chemotherapy, and these findings shed light on a new strategy for DIC prevention.

## 2. Results

### 2.1. Cardiac Injury in Rats after Receiving DOXO (18 mg/kg)

To mimic these clinical settings of DIC, rats received 3 mg/kg of DOXO periodically and reached a cumulative dose at 18 mg/kg within 30 days. At 30 days after the cessation of DOXO treatment, animals were assessed for long-term cardiotoxicity (Figure 1A). To confirm cardiac cell death, apoptotic proteins were measured by Western blot. Results showed that the levels of BCL2-associated X protein (Bax) (Figure 1C) and cleaved-caspase 3 (Figure 1D) in the DOXO-treated group were significantly higher than in the SAL-treated group, while the levels of pro-caspase 3 were similar (Figure 1E). Moreover, sera from rats in DOXO group contained a significantly higher level of NT-proBNP compared to those in the SAL group (Figure 1F). The results from cardiac function measurement show that cardiac functions of rats were impaired after DOXO treatment. At baseline, there were no differences between % LVEF, % LVFS, E/A ratio and LF/HF ratio between SAL and DOXO groups. At endpoint, % LVEF in the DOXO group significantly decreased from baseline (75.80 ± 4.36 vs. 87.86 ± 1.78) and was significantly lower than the SAL group (75.80 ± 4.36 vs. 85.38 ± 3.08) as shown in Figure 1G. Similarly, % LVFS in the DOXO group at endpoint significantly decreased from baseline (39.86 ± 3.70 vs. 52.56 ± 2.45) and was also significantly lower than the SAL group (39.86 ± 3.70 vs. 49.64 ± 3.67) as shown in Figure 1H. These data indicated that left ventricular contractility was impaired in rats at 30 days after receiving DOXO at cumulative dose 18 mg/kg. In addition, we found a significant decrease in E/A ratio at endpoint in the DOXO group when compared to SAL (1.48 ± 0.08 vs. 1.54 ± 0.09) as shown in Figure 1I, which suggests that diastolic function was impaired in the DOXO group. Such a decline in diastolic function of the ventricle is a characteristic of DOXO-induced cardiomyopathy. For cardiac autonomic function, the LF/HF ratio at the endpoint markedly increased in the DOXO group when compared to baseline and the SAL group at the endpoint (0.27 ± 0.05 vs. 0.11 ± 0.02) as shown in Figure 1L, which suggests that DOXO-treated rats had impaired cardiac sympathovagal balance. These results confirmed that the animals developed cardiac injury resulting from DOXO.

### 2.2. Fraction 8 of Serum EV Isolated by SEC as a Proper Fraction for Further Functional Studies

To confirm that the effects on cardiomyocytes and macrophages were due to EVs and not other proteins co-purified with the Evs, SEC was used as a method of choice for our study, and Western blot analysis was used to select the purest fraction for the rest of the experiments. Western blot of tetraspanin proteins—the most commonly used EV markers—such as CD9, CD63 and CD81, revealed that CD63 and CD81 were detectable in the same fractions in both the SAL and DOXO groups. CD63 was detectable in fractions 10–11, while CD81 was detectable in fractions 8–11 (Figure 2B). Luminal endoplasmic reticulum protein (ERp72) was present in fractions 10–11 of specimens from both the SAL and DOXO groups, suggesting ER protein contamination in these two fractions. These data suggested that CD63 and CD81 might not be good markers for EV purity since they were also detectable at the same fractions as ERp72. In contrast, CD9 was undetectable in fractions 10–11, indicating that it is a good EV purity marker. CD9 was strongly detected in fractions 7–8 in the SAL group. However, it was strongly detectable in fraction 8, but weakly in fractions 7 and 9, from the DOXO group (Figure 2B). Since fraction 8 from both the SAL and DOXO groups contained the highest levels of the CD9, this fraction was selected as representative of EVs from SAL (SAL_Evs) and DOXO (DOXO_Evs) animals.

### 2.3. Serum EV Concentrations in DOXO Group Were Higher than in SAL Group

The NTA data of SAL_EVs and DOXO_EVs revealed that the EV size was not different between the two groups. The size of SAL_EVs was 113.48 ± 21.49 nm, while that of DOXO_EVs was 122.35 ± 29.25 nm (Figure 3C). Representative size distribution curves are shown in Figure 3A,B. In contrast, the concentration of EV was significantly higher in the DOXO group (13.13 ± 1.42 × 10^10^ particles/mL) compared to the SAL group (8.71 ± 0.68 × 10^10^ particles/mL) as shown in Figure 3D.

### 2.4. DOXO_EVs Inhibited ROS Production in H_2_O_2_-Treated Cardiomyocytes through Upregulation of Antioxidant Genes

Data from DCFDA assay showed that without H_2_O_2_ treatment, SAL_EVs alone induced cardiomyocytes to produce higher levels of ROS in a dose-dependent manner compared to controls (Figure 4A). However, DOXO_EVs did not induce any change in ROS production. After H_2_O_2_ treatment, the levels of ROS were significantly higher in non-EV-treated cells, as well as SAL_EV-treated cells (Figure 4A). In contrast to SAL_EVs, DOXO_EVs attenuated ROS production in cardiomyocytes treated with H_2_O_2_ as indicated by the similar levels of ROS in the DOXO_EVs group and the control group (Figure 4A).

The ROS generation in H9c2 cells has been complemented with the peroxy orange-1 (PO-1) assay. The data are now shown in (Figure 4B). PO-1 is a monoboronate, cell-permeable fluorescent probe that is specific to H_2_O_2_ [25]. These data are consistent and are also supported by the DCFDA assay result in that DOXO_EVs reduced ROS production in H_2_O_2_-treated cardiomyocytes. 

To examine how DOXO_EVs attenuated ROS production by cardiomyocytes, the antioxidant gene expressions were evaluated. As shown in Figure 4C,D, DOXO_EVs at 10 × 10^9^ particles/mL induced almost 10 times greater expression of both glutathione peroxidase 1 (*Gpx1*) and catalase (*Cat*) genes compared to non-EV-treated cells. SAL_EVs also upregulated both *Gpx1* and *Cat*, but to much lesser extent than DOXO_EVs. In contrast to *Gpx1* and *Cat*, manganese superoxide dismutase (*Sod2*) gene expression levels were suppressed by both SAL_EVs and DOXO_EVs at 10 × 10^9^ particles/mL (Figure 4E). 

### 2.5. DOXO_EVs had More Protective Effect than SAL_EVs in H_2_O_2_-Induced Cardiac Cell Death

To further investigate if the antioxidant effect of DOXO_EVs protected cardiomyocytes from oxidative stress, we assessed cytotoxicity assay in EV pre-treated cardiomyocytes receiving H_2_O_2_. Results showed that both SAL_EVs and DOXO_EVs reduced cell death when the H9c2 cells were exposed to either 100 or 200 µM H_2_O_2_ for 24 h. Of note, the efficacy of DOXO_EVs in attenuating cell death was significantly greater than that of SAL_EVs (Figure 5). 

### 2.6. DOXO_EVs Induced ROS Production and Pro-Inflammatory Gene Expression in Macrophages

Inflammation, especially via the innate immune system, plays an important role in DIC [8,26,27]. Since serum EVs affected ROS production by cardiomyocytes, we speculated that the EVs may affect macrophage ROS production and may modify the inflammatory responses of innate immune cells as well. As shown in Figure 6A, DOXO_EVs (but not SAL_EVs) induced greater ROS production in NR8383 cells, in a dose-dependent manner, compared to that in control cells. By staining with H_2_O_2_-specific probe, PO-1, the data showed that DOXO_EVs at 10 × 10^9^ particles/mL significantly induced NR8383 cells to produce higher H_2_O_2_ production (Figure 6B).

To evaluate whether the macrophages were polarized toward pro-inflammatory (M1) or anti-inflammatory (M2) phenotype after exposure to EVs, we measured pro-inflammatory cytokine gene expression including *Il6* and *Tnf*, as well as anti-inflammatory gene expression including arginase-1 (*Arg1*) and *Tgfb*. RT-PCR data showed that DOXO_EVs at 10 × 10^9^ particles/mL significantly induced at least 10 times higher *Il6* (Figure 6C) and at least 2 times higher *Tnf* (Figure 6D) in pro-inflammatory genes. In contrast, DOXO_EVs suppressed *Arg1* gene (Figure 6E), which encodes an enzyme converting L-arginine to L-ornithine, a process of M2 anti-inflammatory macrophage [28]. Although the *Tgfb1* gene, which encodes an anti-inflammatory cytokine TGF-β, significantly increased when NR8383 cells were treated with SAL_EVs or DOXO_EVs (Figure 6F), there was no difference between SAL_EVs and DOXO_EVs at the same dosage. The results that DOXO_EVs induced *Il6* and *Tnf* and suppressed *Arg1* gene expression suggest that DOXO_EVs promote pro-inflammatory macrophage polarization. 

## 3. Discussion

This study investigated the effects of circulating EVs isolated from serum of rats after the cessation of a 18 mg/kg cumulative dose of DOXO treatment for one month on cardiomyocytes and macrophages. Our animal model mimicked the clinical setting of long-term DIC, in which the patients may develop heart failure years after treatment cessation [29]. Here, we reported an increase in the quantity of EVs after DOXO treatment, as well as differential effects of EVs on redox status of cardiomyocytes and macrophages that promote cardio-protection against DIC.

EVs are released from cells under physiological conditions [30]. EV release serves as a tool for discarding toxic or unnecessary cellular garbage [31], and more importantly, as a vehicle for cell–cell communication [32]. The release of EVs is enhanced by many conditions such as hypoxia and oxidative stress [33]. Cancer treatments, such as radiation and chemotherapy, are known to cause oxidative stress in human tissues. Previous studies have shown that these therapies induced exosome generation and secretion [20,34]. Here, we demonstrated that the oxidative stress that occurs during chronic DOXO treatment persisted and enhanced the secretion of EVs one month after treatment cessation. Our findings are consistent with the previous literature that demonstrates persistent ROS generation in rats five weeks after the last DOXO injection [35]. 

To determine functions of the circulating EVs released after DOXO treatment, the purity of the studied EVs is crucial [36]. This study used SEC for isolation of EVs as a method of choice because it has been demonstrated to be less protein contaminated and less aggregated, and their integrity and biological activities are better preserved compared to EVs collected with other isolation methods [37,38]. In this study, the Western blot analysis showed that fraction 8 of the SEC of SAL and DOXO sera contained high levels of CD9, an EV marker protein that was not present in the later fractions (which contained ERp72, a non-EV protein). Therefore, fraction 8 was selected as the best representative of EVs for the whole study. Since we selected only one fraction of SEC to study, and SEC-isolated EVs based on size, the smaller and larger sizes of EVs in other fractions may be excluded. Therefore, we did not see any difference of size between SAL_EVs and DOXO_EVs. 

Circulating EVs are heterogeneous in their cellular origin and molecular composition and thus in their biological effect on target cells. Flow cytometric analysis of healthy donor peripheral blood reveals that the major sources of EVs are primarily mononuclear phagocytes and platelets [39]. In this study, we found that EVs released under physiological condition (represented by SAL_EVs) induced ROS production in cardiomyocytes. This finding was consistent with work by Gibbins and colleagues, who reported that platelet-derived EVs expressed NADPH oxidase-1 and generated superoxide upon activation [40]. However, despite the induction of ROS production in cardiomyocytes, SAL_EVs were protective against H2O2-induced cell death. Under physiological conditions, ROS are regulators of intracellular signaling pathways by controlling redox-sensitive proteins [41]. Such pathways include growth factor signal transduction [42,43], calcium signaling [44], cell proliferation and differentiation [45]. Therefore, ROS induced by SAL_EVs were sublethal, while the cytoprotective effect of SAL_EVs was mediated by non-antioxidant-related pathway. DOXO_EVs were more effective than SAL_EVs in protecting cardiomyocytes from H2O2-induced cell death. Unlike SAL_EVs, the ability to induce ROS production of DOXO_EVs was attenuated. DOXO_EVs did not cause superoxide detoxification directly since the level of *Sod2* in the cells was not increased. Instead, DOXO_EVs reduced ROS production in cardiomyocytes by inducing of *Gpx1* and *Cat* gene expression—an enzyme in detoxification of hydrogen peroxide— a product of superoxide dismutation. Our data supported the previous literature showing that EVs released under oxidative stress conditions induced anti-oxidative responses in part through the nuclear factor erythroid 2-related factor 2-Kelch-like ECH-associated protein 1 (Nrf2-Keap1) pathway [17,46], the pathway that regulates Gpx1 and Cat gene expression. The cytoprotective effect of DOXO_EVs implied that the cells release EVs to be a compensatory mechanism against oxidative stress. Thus, the concentration of EVs in DOXO-treated patients’ circulation might reflect the ability to tolerate DOXO toxicity. Further study should be performed to support or refute this speculation.

The mechanism of DIC involves not only cardiac cell death, but also systemic immune response. Clinical studies in patients receiving DOXO showed that plasma levels of several cytokines and chemokines are altered in patients with cardiac dysfunction compared to normal subjects [7,47]. In animal studies, although pro-inflammatory gene expression in the myocardium does not increase and may even decrease after DOXO treatment [26,48], systemic elevation in pro-inflammatory cytokines including IL-1β, TNF-α, IL-6, and other chemokines have been reported in many studies [8,49,50]. A recent study revealed that plasma mRNA level of Il6 positively correlated with the severity of DIC and served as a protective factor against DIC [51]. In the present study, we demonstrated a crosstalk between injured cells and immune response by showing that circulating EVs released after DOXO treatment induced ROS production and enhanced Il6 and Tnf gene expression in macrophages. IL-6 is a pleiotropic pro-inflammatory cytokine that plays an important role in the progression of heart failure. Short-term increases in IL-6 promote anti-apoptotic response in myocardium while chronic IL-6 elevation suppresses cardiac contractility [52]. Moreover, IL-6 in the tumor microenvironment promotes tumor progression resistance to chemo- and radiotherapy through the JAK/STAT3 signaling pathway [53,54]. TNFα is also crucial for tumor progression. Low levels of TNFα during chronic inflammation can lead to immune escape and tumor growth by activating immunosuppressive cells such as regulatory T cell, regulatory B cells and myeloid-derived suppressor cells [55]. Therefore, it is possible that EVs released after DOXO treatment can have indirect positive effects on the myocardium and can promote tumor aggressiveness. Further in in vitro study using primary cardiomyocytes instead of the H9c2 cell line, as well as in vivo studies such as EV injection in normal rats or during proinflammatory or oxidative stress, and EV injection in tumor-bearing rats should be performed to confirm our speculations.

## 4. Material and Methods

### 4.1. Ethical Approval

All animal protocols were approved by the Laboratory Animal Center, Chiang Mai University, Chiang Mai, Thailand (approval no. 2562/RT-0008). Animal experiments were conducted following the Guide for the Care and Use of Laboratory Animals.

### 4.2. Animal Preparation, Sera and Tissue Collection

All animal experiments were performed at the Laboratory Animal Center, Chiang Mai University, Chiang Mai, Thailand. Adult male Wistar rats were obtained from the Nomura Siam International Co. ltd, Bangkok, Thailand. The animals were housed in individual ventilated cages with a 12 h light/12 h dark environmental cycle and at a controlled temperature and acclimatized to laboratory conditions for at least one week before starting the experiments. Rats, weighing approximately 300–350 g each, were randomly assigned into the control group or the DOXO-treated group. In DOXO group, rats were injected with 6 doses of doxorubicin (3 mg/kg, ip, ADRIM, Homburg, Germany) on days 0, 4, 8, 15, 22, and 29 as shown in Figure 1. In the control group, rats were intraperitoneally (i.p.) injected with normal saline solution (SAL) in the same volume and treatment schedule as in DOXO group. After the last dose of DOXO or SAL injections for one month, the rats were euthanized by deep anesthesia using an overdose of isoflurane. Under the anesthesia, blood samples were collected from the inferior venae cavae and superior venae cavae (to avoid traumatic cardiac injury). Left ventricles were collected and washed with normal saline, then snap-frozen with liquid nitrogen. All of heart tissues were kept at −80 °C until used. Blood was allowed to clot at room temperature for 30 min. Then, the clotted blood was centrifuged at 1500× *g* for 5 min, and the supernatants were collected. To remove platelets, the supernatants were further centrifuged at 3000× *g* for 15 min. The final supernatants were collected without pellet disturbance and were kept at −80 °C until used.

### 4.3. Cardiac Function Measurement

Prior to treatment, cardiac functions were evaluated in all rats by echocardiography and heart rate variability (HRV). The values were recorded as baseline cardiac functions. During echocardiography, the rats were anesthetized under 2% isoflurane with oxygen flow 2 L/min. Echocardiographic parameters were measured using a Vivid*i (GE Healthcare, UK). The echocardiography probe was placed in gentle contact with the chest, and images were collected along the parasternal short axis of the heart. M-mode echocardiography was performed at the level of the papillary muscles. Then, % fractional shortening and left ventricular ejection fraction were determined. In addition, the apical four-chamber view was obtained to evaluate a pulsed-wave Doppler spectrum of transmitral flow. Transmitral early filling to atrial filling velocities (E/A) ratio was determined to indicate the diastolic function [56]. All echocardiographic parameters were measured again at day 60 after the first dose of doxorubicin or saline injection and compared cardiac functions to baseline. 

HRV was performed at baseline and endpoint to evaluate sympathovagal balance of the animals. Lead II electrocardiogram was recorded using a Chart 5.0 program and a PowerLab 4/25T (ADInstruments, Inc., Sydney, Australia). A selected section of the tachogram of at least 300 consecutive RR intervals was evaluated as described previously [56]. The high-frequency component (HF) represented the parasympathetic tone. The low-frequency component (LF) represented the parasympathetic and sympathetic activities. The LF/HF ratio was determined as an indicator of sympathovagal balance. Parasympathetic withdrawal and/or sympathetic overactivity were indicated by an increased LF/HF ratio [57] (Figure 1A).

### 4.4. EV Isolation

EVs were isolated by a size-exclusion chromatography (SEC) technique using a qEVoriginal-70 nm column (Izon Science, New Zealand). The procedure was performed according to the protocol provided by the company with some modification. In brief, sera were thawed at room temperature, then centrifuged at 10,000× *g* for 10 min. Before loading the samples, columns were flushed with at least 15 mL of 0.22 μm-filtered PBS. Centrifuged sera (500 μL each) were loaded onto the loading frit. Once all of the sera entered the column, filtered PBS (10 mL) was used to flush the columns. The flow-through fluid was collected from fractions 1 (F1) to 11 (F11), 500 μL per each fraction. Fractions F1 to F4 (2 mL) were voided, while F5 to F11 was stored at −80 °C before use in the following experiments as shown in Figure 2A. All of the isolation steps were performed in a cell culture hood. 

### 4.5. Western Blot Analysis

For EV marker detection, equal volumes (500 μL) of each fraction from F5 to F11 were concentrated ten-fold using Amicon^®^ Ultra-0.5 mL 3K centrifugal filter units (MilliporeSigma, Burlington, MA, USA). Starting with 500 μL of each eluted fraction, the fluid was centrifuged for 30 min at 14,000× *g*, room temperature to obtain 50 μL of concentrate. Then, the concentrated samples were mixed with 6× loading buffer and boiled at 95 °C. After that, the mixture was loaded into 12% sodium dodecyl sulfate-polyacrylamide gel, and electrophoresis (SDS-PAGE) was performed at 60 V for 30 min, then at 100 V for 1 h. The protein was transferred to a PVDF membrane (MilliporeSigma, Burlington, MA, USA) in a Western blot transfer system (Bio-Rad, Hercules, CA, USA). The membranes were then blocked with 5% non-fat dry milk in Tris Buffered Saline with Tween for 1 h at room temperature. After blocking, the membranes were incubated overnight at 4 °C with anti-CD63 (1:1000), anti-CD9 (1:1000), anti-CD81 (1:1000) using ExoAb Antibody Kit (System Biosciences, Palo Alto, CA, USA, catalog number EXOAB-KIT-1), and anti-ERp72 at a concentration 1:1000 dilution (Cell Signaling Technology, Danvers, MA, USA, catalog number 5033T). After washing step, the membranes were incubated with anti-rabbit antibody (System Biosciences, Palo Alto, CA, USA) for 1 h at room temperature. The protein bands were visualized by incubating with enhanced chemiluminescence detection reagents (Bio-Rad, USA) using ChemiDoc^TM^ XRS+ Imaging System (Bio-Rad, Hercules, CA, USA). 

For heart tissue, left ventricles were homogenized in buffer containing 20 mM Tris HCl (pH 6.8), 1 mM Na_3_VO_4_, 5 mM NaF, and 1× protease inhibitor. Tissue protein concentrations were measured using Pierce^TM^ BCA Protein Assay Kit (ThermoFisher Scientific, Waltham, MA, USA). Equal amounts of protein (50 μg) from each sample were loaded into 12% SDS-PAGE gel. The process of Western blotting followed the above-mentioned protocol. Anti-Bax at a concentration 1:1000 dilution (Cell Signaling Technology, Danvers, MA, USA, catalog number 2772T), Anti-pro- and cleaved caspase-3 at a concentration 1:1000 dilution (Cell Signaling Technology, Danvers, MA, USA, catalog number 9661S) and Anti-GAPDH at a concentration of 1:1000 dilution (Cell Signaling Technology, Danvers, MA, USA, catalog number 5174T) were used as primary antibodies.

### 4.6. Serum Cardiac Injury Biomarker Measurement

An ELISA assay kit (Elabsciences, Houston, TX, USA) was used to measure the N-terminal pro b-type natriuretic peptide (NT-proBNP) levels in rat serum. Briefly, 100 μL serum was added into a 96-well-microplate pre-coated with anti-rat NT-proBNP antibody and was incubated at 37 °C for 90 min. After removing the liquid from each well, the biotinylated detection antibody was added and incubated at 37 °C for 1 h. Once finishing the washing step, an HRP conjugate solution was added and incubated at 37 °C for 30 min, and the substrate reagent was applied and incubated at 37 °C for 15 min followed by adding a stop solution. The absorbance at 450 nm was measured using a microplate reader (BioTek, Winooski, VT, USA).

### 4.7. Nanoparticle Tracking Analysis (NTA)

EV concentration and size were analyzed using a NanoSight NS300 (Malvern Panalytical, Malvern, UK) equipped with a 488 nm laser. EV suspensions were diluted (1:200–1:400) in filtered PBS. Samples were analyzed under constant flow conditions (flow rate: 30) at 25 °C and were captured with a camera level of 13–14 using NanoSight NTA software version 3.4 (Malvern Pana- lytical). Five independent measurements (60 s each) were obtained for each sample. Data are reported as mean ± standard error of mean (SE). 

### 4.8. Cell Culture

H9c2 rat cardiomyocyte and NR8383 rat macrophage cell lines were obtained from the American Type Culture Collection (Manassas, VA, USA). H9c2 cells were maintained in DMEM low glucose supplemented with 10% FBS, 1% penicillin/streptomycin, and 2 mM L-glutamine. NR8383 cells were maintained in Ham’s F12K medium supplemented with 15% heat inactivated FBS, 2 mM L-glutamine, 1% penicillin/streptomycin, and 1.5 g/L sodium bicarbonate. 

### 4.9. LDH Cytotoxicity Assay

H9c2 cells were plated onto 96-well plates at a concentration of 5000 cells/well. The next day, the medium was changed to the H9c2 growing medium supplemented with 10% exosome depleted FBS (System Biosciences, Palo Alto, CA, USA) instead of the regular FBS (EV-free medium). Then, the cells were treated with 10 × 10^9^ or 20 × 10^9^ particles/mL of serum EVs derived from saline-treated rats (SAL_EVs) or serum EVs derived from DOXO-treated rats (DOXO_EVs). The cells were incubated with EVs for 24 h. After that, the cells were washed with PBS and treated with 100 or 200 μM hydrogen peroxide (H_2_O_2_) in EV-free medium for 24 h. The maximum LDH release was induced by adding 10 μL of 10X Lysis Buffer to the cells plated at the same concentration without any treatment and incubated the cells for 45 min in an incubator at 37 °C. After incubation, cell death was evaluated using a CyQUANT^TM^ LDH Cytotoxicity Assay Kit (Invitrogen, Waltham, MA, USA). Briefly, the supernatants of the treated cells were collected and mixed with the reaction mixture. After incubation for 30 min, a stop solution was applied. The absorbance at 490 and 680 nm was measured. Percentage of cytotoxicity was calculated using the following formula: % cytotoxicity = [ (H_2_O_2_-treated LDH activity—spontaneous LDH activity)/(maximum LDH activity—spontaneous LDH activity)] × 100. 

### 4.10. ROS Measurement

H9c2 cells were seeded into black 96-well plates (SPL Life Sciences, Gyeonggi-do, Korea) to achieve a concentration of 5000 cells/well. Then, the cells were incubated with 5 × 10^9^ or 10 × 10^9^ particles/mL of SAL_EVs or DOXO_EVs suspended in EV-free medium for 24 h. Next, the cells were treated with PBS or 100 μM H_2_O_2_ for 10 min and were washed with PBS and measured ROS production using 2′,7′-dichlorofluorescin acetate or DCFDA (Sigma-Aldrich, St. Louis, MO, USA). The cells were incubated with 25 μM DCFDA for 20 min and were washed with PBS; then, the fluorescence intensity was assessed using a fluorescence microplate reader with excitation and emission wavelengths at 485 and 535 nm, respectively.

NR8383 cells at a concentration 25,000 cells/well were seeded onto black 96-well plates. Then, the cells were incubated with 5 × 10^9^ or 10 × 10^9^ particles/mL of SAL_EVs or DOXO_EVs suspended in EV-free medium for 24 h. The ROS measurement was performed as above.

### 4.11. H_2_O_2_ Measurement

H9c2 cells were seeded into black 96-well plates to achieve a concentration of 5000 cells/well. Then, the cells were incubated with 5 × 10^9^ or 10 × 10^9^ particles/mL of SAL_EVs or DOXO_EVs suspended in EV-free medium for 24 h. Next, the cells were treated with PBS or 100 μM H_2_O_2_ for 10 min and were washed with PBS. Cytosolic H_2_O_2_ levels were measured by using peroxy orange-1 (PO-1) as a fluorescent probe. PO-1 is a monoboronate, cell-permeable fluorescent probe that is specific to H_2_O_2_ [25]. The fluorescence intensities were assessed using fluorescence spectroscopy with excitation/emission at 485 nm/535 nm using a fluorescence microplate reader. 

NR8383 cells at a concentration 10,000 cells/well were seeded onto black 96-well plates. Then, the cells were incubated with 5 × 10^9^ or 10 × 10^9^ particles/mL of SAL_EVs or DOXO_EVs suspended in EV-free medium for 24 h. The H_2_O_2_ measurement was performed using PO-1 as above.

### 4.12. Gene Expression Measurement

H9c2 cells at a concentration 1 × 10^5^ cells/well or NR8383 cells at a concentration 1 × 10^6^ cells/well were seeded into 6-well plates. The next day, the cells were treated with EV-free media supplemented with 5 × 10^9^ or 10 × 10^9^ particles/mL of SAL_EVs or DOXO_EVs suspended in EV-free medium for 24 h. After 24 h, the cells were collected by cell scraping and kept in −80 °C until used.

RNA was isolated using RNeasy mini kit (Qiagen, Hilden, Germany). Complementary DNA (cDNA) was synthesized from 1 μg of total RNA by reverse transcription using iScript cDNA synthesis kit (Bio-Rad, Hercules, CA, USA). Concentration of cDNA was measured by NanoDrop^TM^ 2000/2000c Spectrophotometers (ThermoFisher Scientific, Waltham, MA, USA). Then, 100 ng/μL of cDNA was prepared in RNase-free water and gene expression levels were determined by quantitative real-time PCR using CFX96 Touch real-time PCR detection system (Bio-Rad, Hercules, CA, USA). Primer sequences for rat genes were as listed in Table 1. The gene expression levels were calculated by the 2^−ΔΔCt^ method and the data were presented as fold change relative to cells treated with PBS.

### 4.13. Statistical Analysis

Results were summarized as mean ± standard deviation (SD) or standard error of mean (SEM), as mentioned in the figure legend. Student *t* tests were used to analyze the differences between findings in the SAL and DOXO groups. Analysis of variance (ANOVA) was used to compare the differing doses of EV treatment with each parameter. Tukey’s multiple comparisons test was used in order to determine differences between groups after ANOVA. *p* value < 0.05 was considered significant. Excel and GraphPad Prism version 6.0c were used for data analysis and graphing.

## 5. Conclusions

EVs in the circulation after cessation of DOXO treatment were increased in number and were also biologically active. Functional study of these EVs suggested that oxidative stress conditions might alter the molecular cargo of EVs, leading to the ability to modulate redox status of the target cells including cardiomyocytes and macrophages. Their effects on redox status and the gene expression of cardiomyocytes and macrophages were distinct. Thus, the net consequences of EVs on cancer patients remain unclear regarding whether they are more protective or detrimental against DIC and tumor progression. Further studies of the molecular contents of DOXO_EVs, as well as in vivo studies of DOXO_EVs in tumor-bearing animals, may be needed to further decipher the mechanisms and functions of EVs. Such studies may lead to novel approaches to the prevention of DOXO-induced cardiomyopathy.

## Figures and Tables

**Figure 1 ijms-23-13465-f001:**
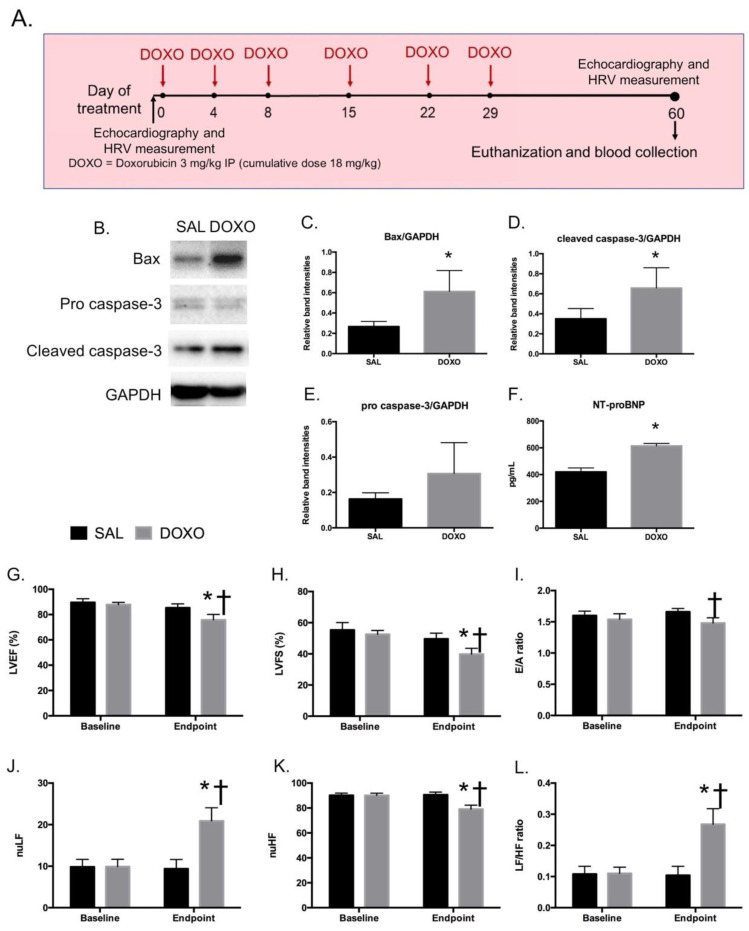
Timeline of DOXO treatment and blood collection in Wistar rats (**A**). Representative picture of Western blot of Bax, cleaved caspase-3, pro caspase-3, and GAPDH from heart tissues of saline-treated (SAL) and DOXO-treated (DOXO) rats (**B**). Fold change of band intensities of Bax relative to GAPDH (**C**), cleaved caspase-3 relative to GAPDH (**D**), pro caspase-3 relative to GAPDH (**E**) in SAL (*n* = 5) and DOXO (*n* = 5) groups. Concentration of NT-proBNP in serum of SAL (*n* = 5) and DOXO (*n* = 5) groups (**F**); * *p* < 0.05 vs. SAL. The effect of Doxorubicin on left ventricular function and cardiac sympathovagal balance (**G**–**L**). (**G**). LVEF (%), (**H**). LVFS (%), (**I**). E/A ratio, (**J**). LF (normalized unit), (**F**). HF (normalized unit), (**G**). LF/HF ratio; DOXO, doxorubicin; LVEF, left ventricular ejection fraction; LVFS, left ventricular fractional shortening, E/A, early filling to atrial filling velocities ratio; LF/HF, low-frequency component to high-frequency component ratio. Data are presented as the mean ± SEM; * *p* < 0.05 vs. baseline; † *p* < 0.05 vs. SAL at that period of time.

**Figure 2 ijms-23-13465-f002:**
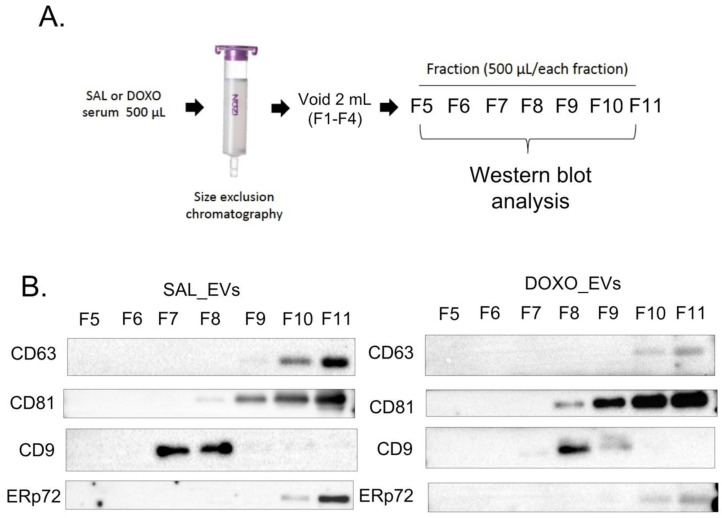
Diagram illustrating isolation of serum Evs using size exclusion chromatography (SEC) including void volume and collection of flow though fractions for Western blot analysis (**A**), Western blot of potential EV markers (CD63, CD9, and CD81), as well as a non-EV protein—ERp72—in fractions 5 to 11 from SEC of SAL and DOXO-rat sera (**B**).

**Figure 3 ijms-23-13465-f003:**
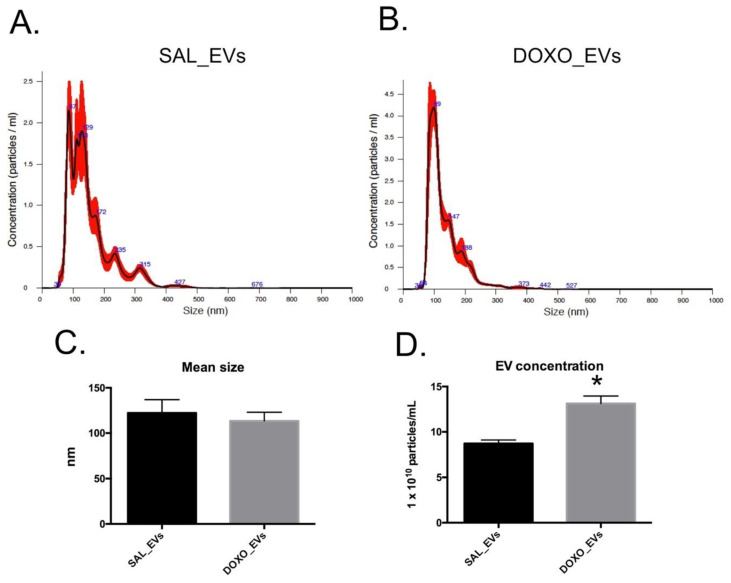
Representative graphs of size distributions of SAL_EVs (**A**) and DOXO_EVs (**B**) size distribution. Blue line represents the averaged data, whereas red line represents the SE. The mean size ± SE of SAL_EVs and DOXO_EVs (**C**)The average EV concentration ± SE of SAL_EVs (*n* = 5) and DOXO_EVs (*n* = 5) (**D**). * *p* < 0.05.

**Figure 4 ijms-23-13465-f004:**
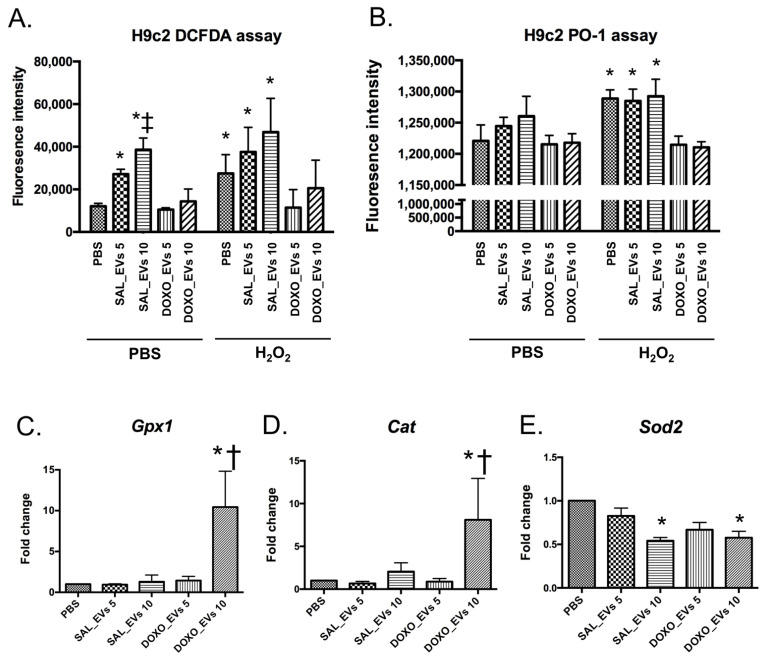
Levels of ROS production measured by DCFDA assay (**A**) and PO-1 assay (**B**) in H9c2 cells pre-incubated with SAL_EVs or DOXO_EVs for 24 h followed by PBS or 100 μM H_2_O_2_ treatment for 10 min; * *p* < 0.05 vs. no EV, no H_2_O_2_ treated control (PBS), ‡ *p* < 0.05 vs. SAL_EVs 5 Levels of *Gpx1* (**C**), *Cat* (**D**) and *Sod2* (**E**) gene expression in H9c2 cells incubated with SAL_EVs or DOXO_EVs for 24 h; SAL_EVs 5 = H9c2 cells pre-incubated with SAL_EVs 5 × 10^9^ particles/mL; SAL_EVs 10 = H9c2 cells pre-incubated with SAL_EVs 10 × 10^9^ particles/mL; DOXO_EVs 5 = H9c2 cells pre-incubated with DOXO_EVs 5 × 10^9^ particles/mL; DOXO_EVs 10 = H9c2 cells pre-incubated with DOXO_EVs 10 × 10^9^ particles/mL; * *p* < 0.05 vs EVs from the same group at concentration 5 × 10^9^ particles/mL, † *p* < 0.05 vs SAL_EVs 10 × 10^9^ particles/mL.

**Figure 5 ijms-23-13465-f005:**
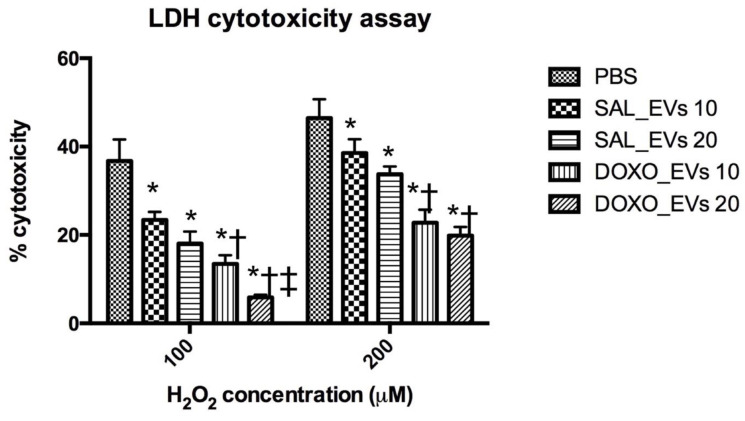
Percentage of cell death measured by LDH cytotoxicity assay in H9c2 cells pre-incubated with PBS, SAL_EVs or DOXO_EVs followed by 100 or 200 μM H_2_O_2_ exposure for 24 h. SAL_EVs 10 = H9c2 cells pre-incubated with SAL_EVs 10 × 10^9^ particles/mL, SAL_EVs 20 = H9c2 cells pre-incubated with SAL_EVs 20 × 10^9^ particles/mL; DOXO_EVs 10 = H9c2 cells pre-incubated with DOXO_EVs 10 × 10^9^ particles/mL; DOXO_EVs 20 = H9c2 cells pre-incubated with DOXO_EVs 20 × 10^9^ particles/mL; * *p* < 0.05 vs. no EV treated control (PBS), † *p* < 0.05 vs. SAL_EVs 10 × 10^9^ particles/mL; ‡ *p* < 0.05 vs. DOXO_EVs 10 × 10^9^ particles/mL.

**Figure 6 ijms-23-13465-f006:**
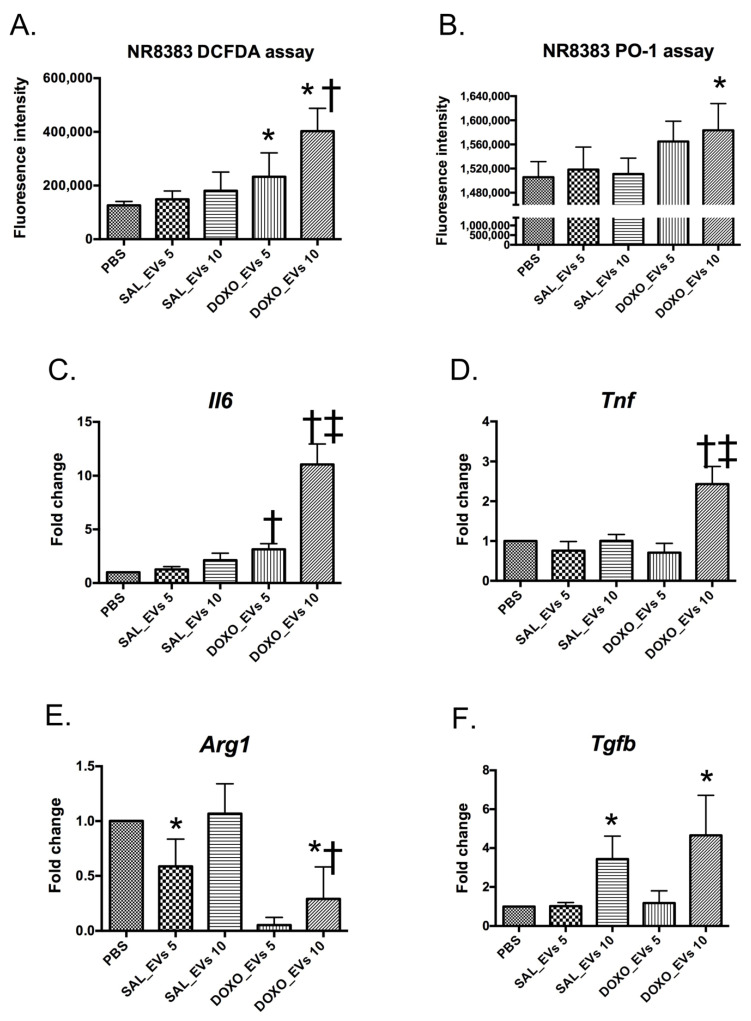
Levels of ROS production measured by DCFDA assay (**A**) and PO-1 assay (**B**) in NR8383 cells pre-incubated with SAL_EVs or DOXO_EVs for 24 h; SAL_EVs 5 = NR8383 cells pre-incubated with SAL_EVs 5 × 10^9^ particles/mL; SAL_EVs 10 = NR8383 cells pre-incubated with SAL_EVs 10 × 10^9^ particles/mL; DOXO_EVs 5 = NR8383 cells pre-incubated with DOXO_EVs 5 × 10^9^ particles/mL; DOXO_EVs 10 = NR8383 cells pre-incubated with DOXO_EVs 10 × 10^9^ particles/mL; * *p* < 0.05 vs. no EV (PBS), † *p* < 0.05 vs. DOXO_EVs 5 × 10^9^ particles/mL. Levels of *Il6* (**C**), *Tnf* (**D**), *Arg1* (**E**), *Tgfb* (**F**) gene expression in NR8383 cells incubated with SAL_EVs or DOXO_EVs for 24 h; * *p* < 0.05 vs. PBS, † *p* < 0.05 vs. SAL_EVs 5 × 10^9^ particles/mL, ‡ *p* < 0.05 vs. DOXO_EVs 10 × 10^9^ particles/mL.

**Table 1 ijms-23-13465-t001:** List of primer sequences used in real-time RTPCR.

Gene Name	Forward Primer	Reverse Primer
*Gpx1*	5′-AGTTCGGACATCAGGAGAATGGCA-3′	5′-TCACCATTCACCTCGCACTTCTCA-3′
*Cat*	5′-GCACTACAGGCTCCGAGATGAAC-3′	5′-TTGTCGTTGCTTGGTTCTCCTTGT-3′
*Sod2*	5′-AACGTCACCGAGGAGAAGTA-3′	5′-TGATAGCCTCCAGCAACTCT-3′
*Il6*	5′-TCCTACCCCAACTTCCAATGCTC-3′	5′-TTGGATGGTCTTGGTCCTTAGCC-3′
*Tnf*	5′-CCAGGAGAAAGTCAGCCTCCT-3′	5′-TCATACCAGGGCTTGAGCTCA-3′
*Tgfb1*	5′-CTTCAGCTCCACAGAGAAGAACTGC-3′	5′-CACGATCATGTTGGACAACTGCTCC-3′
*Arg1*	5′-CCTGAAGGAACTGAAAGGAAAG-3′	5′-TTGGCAGATATGCAGGGAGT-3′
*Actb*	5′-CACTGGCATTGTGATGGACT-3′	5′-CTCTCAGCTGTGGTGGTGAA-3′

## Data Availability

Data are available on reasonable request to the corresponding author.

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
