# Peer review of "Extracellular Vesicles Released after Doxorubicin Treatment in Rats Protect Cardiomyocytes from Oxidative Damage and Induce Pro-Inflammatory Gene Expression in Macrophages"

_ijms, 2022, doi:10.3390/ijms232113465_

Round 1

Reviewer 1 Report

The manuscript of Yarana et al.  presents data suggesting that extracellular vesicles (EVs), isolated from the serum of rat undergone DOXO treatment, prevent oxidative stress in H9c2 cells, while inducing ROS production and pro-inflammatory genes in macrophages.

Although the topic of the manuscript is of scientific interest in the cardiovascular field, experimental research is not well designed, and conclusions are not well supported by the results presented in the current version of the manuscript.

The major comments of this reviewer are:

1)    The assessment that DOXO treatment induces DOXO-induced cardiomyopathy (DIC) requires functional data of heart contractile dysfunction and histopathological analysis of the heart, with a focus on tissue inflammation, activation of oxidative stress and myocardial remodeling;

2)    In vitro experiments evaluating oxidative stress and macrophage activation requires appropriate controls. Furthermore, the authors need to improve the analyses used to assess the state of cellular oxidative stress and inflammatory cell activation;

3)    Rat neonatal cardiomyocytes (CMs) would be used, instead of H9c2, to assess the effects of DOXO-derived EVs on cell oxidative stress.;

4)    In vitro assays need to be confirmed by in vivo experiments. As an example, what about the effects of EVs, from DOXO-treated rats, injected in normal rats at baseline or during proinflammatory/oxidative stresses?

Author Response

We would like to thank the reviewer #1 for his/her valuable comments on the manuscript. To address the concerns raised, new figures have been added. Changes to the text are tracked in the revised manuscript.

1. The assessment that DOXO treatment induces DOXO-induced cardiomyopathy (DIC) requires functional data of heart contractile dysfunction and histopathological analysis of the heart, with a focus on tissue inflammation, activation of oxidative stress and myocardial remodeling;

         Answer: We would like to thank the reviewer for this useful suggestion. In the revised manuscript, we include the functional data of heart contractile dysfunction in Figure 1G-1L. The results show that cardiac functions of rats were impaired after DOXO treatment. At baseline, there were no differences between % LVEF, % LVFS, E/A ratio and LF/HF ratio between SAL and DOXO groups. At endpoint, % LVEF in DOXO group significantly decreased from baseline (75.80 ± 4.36 vs. 87.86 ± 1.78) was significantly lower than SAL group (75.80 ± 4.36 vs. 85.38 ± 3.08) as shown in Figure 1G. Similarly, % LVFS in DOXO group at endpoint significantly decreased from baseline (39.86 ± 3.70 vs 52.56 ± 2.45) and was also significantly lower than SAL group (39.86 ± 3.70 vs 49.64 ± 3.67) as shown in Figure 1H. These data indicated that left ventricular contractility was impaired in rats at 30 days after receiving DOXO at cumulative dose 18 mg/kg. In addition, we found a significant decrease in E/A ratio at endpoint in DOXO group when compared to SAL (1.48 ± 0.08 vs 1.54 ± 0.09) as shown in Figure 1I, which suggests that diastolic function was impaired in DOXO group. Such decline in diastolic function of ventricle is a characteristic of DOXO-induced cardiomyopathy. For cardiac autonomic function, LF/HF ratio at the endpoint markedly increased in DOXO group when compared to baseline and SAL group at the endpoint (0.27 ± 0.05 vs 0.11 ± 0.02) as shown in Figure 1L, which suggests that DOXO-treated rats had impaired cardiac sympathovagal balance. Cardiac function measurement method can be found in the revised manuscript on page 3, paragraph 2-3, lines 114-136. The cardiac function results can be found on page 6, paragraph 4, lines 284-287 and page 7, paragraph 1, lines 288-301.

Although we do not show the histopathological data demonstrating cardiac tissue inflammation and remodeling in this study, our group has reported in a previous study that rats that received DOXO treatment in the same treatment protocol had elevated levels of pro-inflammatory cytokines gene expression including TNF alpha and IL-6 in cardiac tissues. Moreover, in the same study, we have also reported the activation of oxidative stress as evidenced by increased lipid peroxidation product—malondialdehyde (MDA) —in cardiac tissues of DOXO-treated rats [1].

2. In vitro experiments evaluating oxidative stress and macrophage activation requires appropriate controls. Furthermore, the authors need to improve the analyses used to assess the state of cellular oxidative stress and inflammatory cell activation;

Answer: In our in vitro experiments of macrophage activation, the negative controls for DCFDA and RTPCR assays were NR8383 cells treated with PBS, which was the buffer that we used to resuspend EVs. However, we did not include positive controls for DCFDA and RTPCR assays because EVs already had positive effect on these assays.

We have done additional analysis based on the reviewer suggestion.  In the revised manuscript, the ROS generation in the cellular models, both H9c2 and NR8383 cells, has been complemented with the peroxy orange-1 (PO-1) assay. The data are now shown in Figure 4B and Figure 6B. PO-1 is a monoboronate, cell-permeable fluorescent probe that is specific to hydrogen peroxide [2]. The new data are consistent and are also supported the DCFDA assay result in that DOXO_EVs reduced ROS production in H2O2-treated cardiomyocytes and induced ROS production in macrophages.  PO-1 method was described in the revised manuscript on page 5, paragraph 5-6, lines 234-246. The result was added on page 10, paragraph 2, lines 357-361 for H9c2 cells and on page 12, paragraph 1, lines 399-401 for NR8383 cells.

3. Rat neonatal cardiomyocytes (CMs) would be used, instead of H9c2, to assess the effects of DOXO-derived EVs on cell oxidative stress.; 

Answer: We would like to thank the reviewer for this very useful suggestion. Rat neonatal cardiomyocytes, which are primary cells that retains physiological functions should also be used to assess the effects of DOXO-derived EVs on cell oxidative stress. However, using such primary cell culture is a limitation of our study due to animal ethical issues, which takes long time to get approval from the ethical committees.

4. In vitroassays need to be confirmed by in vivo As an example, what about the effects of EVs, from DOXO-treated rats, injected in normal rats at baseline or during proinflammatory/oxidative stresses? 

Answer: We think this is an excellent suggestion. We agree that this is a potential limitation of this study. Further investigations to explore the effects of EVs in vivo should be done to confirm our in vitro data. We have added this idea in the revised manuscript on page 14, paragraph 2, lines 1-5 for H9c2 cells and on page 15, paragraph 2, lines 503-505.  

Reference:

  1. Arinno, A.; Maneechote, C.; Khuanjing, T.; Ongnok, B.; Prathumsap, N.; Chunchai, T.; Arunsak, B.; Kerdphoo, S.; Shinlapawittayatorn, K.; Chattipakorn, S. C.; Chattipakorn, N., Cardioprotective effects of melatonin and metformin against doxorubicin-induced cardiotoxicity in rats are through preserving mitochondrial function and dynamics. Biochem Pharmacol 2021, 192, 114743.
  2. Dickinson, B. C.; Huynh, C.; Chang, C. J., A palette of fluorescent probes with varying emission colors for imaging hydrogen peroxide signaling in living cells. J Am Chem Soc 2010, 132, (16), 5906-15.

Reviewer 2 Report

The manuscript describes the effect of extracellular vesicles (EVs) obtained from serum from rats exposed to DOX up to 1 month prior to the serum collection on the redox status of cardiomyocytes and macrophage differentiation. The most interesting aspect of the manuscript is that EVs obtained a month after completion of the treatment with DOX still produce different effects.

On the other hand, the translational relevance is rather limited since effects are not only specific of DOX-EVs (control EVs partially trigger similar effects). In addition, data points to a cytoprotective effect of DOX-EVs on cardiomyocytes but, simultaneously, the same EVs seem to stimulate macrophage polarization to a pro-inflammatory M1 phenotype.  

The following aspects need to be specifically addressed:

1.       Line 121. According to the manuscript, the column was flushed was 10 mL filtered PBS and 11 fractions of 500 uL each were collected. Altogether, this yields a volume of 5.5 mL (not 10 mL). Please, clarify.

2.       Western blot analysis. Please, indicate the centrifugation conditions (time, speed, temperature) used for the concentration of the EVs described in lines 127-130.

3.       Western blot analysis. Please, indicate the catalogue number and dilutions of all the primary antibodies used for western blot.

4.       Did the authors used the same antibody to detect both pro-caspase 3 and cleaved-caspase 3? Please, specify.

5.       LDH cytotoxicity assay. It is not clear whether the EVs were still present in the treatment medium during the treatment with H2O2. Please, specify in materials and methods.

6.       LDH cytotoxicity assay. How did the authors determine the maximum LDH activity?

7.       Statistical analysis. Which post-test was used in order to determine differences between groups after ANOVA?

Author Response

We would like to thank the reviewer #2 for raising the critical points. To respond to the reviewer's concern, we have added new information in the revised manuscript.

  1. Line 121. According to the manuscript, the column was flushed was 10 mL filtered PBS and 11 fractions of 500 uL each were collected. Altogether, this yields a volume of 5.5 mL (not 10 mL). Please, clarify.

Answer: Thank you for pointing this out. We have collected the fractions later than fraction 11th to reach 10 mL of the flushed volume and detected ERp72 by western blot analysis. However, all of those fractions contained ERp72, which indicates the non-EV protein contamination. Therefore, those fractions were excluded from our study.

  1. Western blot analysis. Please, indicate the centrifugation conditions (time, speed, temperature) used for the concentration of the EVs described in lines 127-130.

Answer: Thank you for pointing this out. Starting with 500 of each eluted fraction, the fluid was centrifuged for 30 minutes at 14,000 x g, room temperature to be concentrated to 50 . The centrifugation conditions used for the concentration of the EVs have been added on the revised manuscript on page 4, paragraph 1, lines 152-153.

  1. Western blot analysis. Please, indicate the catalogue number and dilutions of all the primary antibodies used for western blot.

Answer: Thank you for the useful suggestion. We used anti-CD63 (1:1,000), anti-CD9 (1:1,000), anti-CD81 (1:1,000) from ExoAb Antibody Kit (System Biosciences, USA, catalog number EXOAB-KIT-1), and anti-ERp72 at a concentration 1:1,000 dilution from Cell Signaling Technology, USA, catalog number 5033T for EV western blot. Anti-Bax at a concentration 1:1,000 dilution (Cell Signaling Technology, USA, catalog number 2772T), Anti-pro- and cleaved caspase-3 at a concentration 1:1,000 dilution (Cell Signaling Technology, USA, catalog number 9661S) and Anti-GAPDH at a concentration 1:1,000 dilution (Cell Signaling Technology, USA, catalog number 5174T) were used as primary antibodies for cardiac tissue western blot. The details of all primary antibodies used for western blot was indicated in the revised manuscript on page 4, paragraph 1, lines 160-163, and on paragraph 2, lines 172-176.

  1. Did the authors used the same antibody to detect both pro-caspase 3 and cleaved-caspase 3? Please, specify.

Answer: Yes, we used anti-pro- and cleaved caspase-3 at a concentration 1:1,000 dilution (Cell Signaling Technology, USA, catalog number 9661S) as the same antibody to detect both pro-caspase 3 and cleaved-caspase 3.

  1. LDH cytotoxicity assay. It is not clear whether the EVs were still present in the treatment medium during the treatment with H2O2. Please, specify in materials and methods.

Answer: Thank you for concerning a very important point. The cells were incubated with EVs for 24 hr. After that, the cells were washed with PBS and treated with 100 or 200 M hydrogen peroxide (H2O2) in EV-free medium for 24 hr. Therefore, the EVs were not present in the treatment medium during H2O2 treatment. We have added this information in the revised manuscript on page 5, paragraph 2, lines 208-210.

  1. LDH cytotoxicity assay. How did the authors determine the maximum LDH activity?

Answer: The maximum LDH release was induced by adding 10 L of 10X Lysis Buffer to the cells plated at the same concentration without any treatment and incubated the cells for 45 minutes in an incubator at 37 .  We have added this information in the revised manuscript on page 5, paragraph 2, lines 210-212.

  1. Statistical analysis. Which post-test was used in order to determine differences between groups after ANOVA?

Answer: We would like to apologize for the missing information. Tukey's multiple comparisons test was used in order to determine differences between groups after ANOVA. We have added this information in the revised manuscript on page 6, paragraph 3, lines 270-271.

Round 2

Reviewer 2 Report

The authors have addressed all my concerns. I recommend the publication of the manuscript.

Author Response

We sincerely thank the reviewer for doing an in-depth analysis and giving us such useful suggestion.